# Biological Activity of an *Epilobium angustifolium* L. (Fireweed) Infusion after In Vitro Digestion

**DOI:** 10.3390/molecules27031006

**Published:** 2022-02-02

**Authors:** Klaudia Kowalik, Magdalena Polak-Berecka, Monika Prendecka-Wróbel, Dominika Pigoń-Zając, Iwona Niedźwiedź, Dominik Szwajgier, Ewa Baranowska-Wójcik, Adam Waśko

**Affiliations:** 1Department of Biotechnology, Microbiology and Human Nutrition, University of Life Sciences in Lublin, Skromna 8, 20-704 Lublin, Poland; klaudia.kowalik@up.lublin.pl (K.K.); koprukowniak.i@gmail.com (I.N.); dominik.szwajgier@up.lublin.pl (D.S.); ewa.baranowska@up.lublin.pl (E.B.-W.); adam.wasko@up.lublin.pl (A.W.); 2Chair and Department of Human Physiology, Medical University of Lublin, Radziwiłłowska 11, 20-080 Lublin, Poland; monikaprendeckawrobel@umlub.pl (M.P.-W.); dominika.pigon-zajac@umlub.pl (D.P.-Z.)

**Keywords:** colon cancer, *Epilobium angustifolium*, in vitro digestion, polyphenols, plant extracts, ECIS, cytotoxicity, impedance

## Abstract

The biological activity of an in vitro digested infusion of *Epilobium angustifolium* (fireweed) was examined in a model system of intestinal epithelial and colon cancer tissues. The content of selected phenolic compounds in the digested aqueous extract of fireweed was determined using HPLC-ESI-QTOF-MS/MS. Biological activity was examined using the human colon adenocarcinoma cell lines HT-29 and CaCo-2 and the human colon epithelial cell line CCD 841 CoTr. Cytotoxicity was assessed by an MTT assay, a Neutral Red uptake assay, May-Grünwald-Giemsa staining, and a label-free Electric Cell-Substrate Impedance Sensing cytotoxicity assay. The effect of the infusion on the growth of selected intestinal bacteria was also examined. The extract inhibited the growth of intestinal cancer cells HT-29. This effect can be attributed to the activity of quercetin and kaempferol, which were the most abundant phenolic compounds found in the extract after in vitro digestion. The cytotoxicity of the fireweed infusion was dose-dependent. The highest decrease in proliferation (by almost 80%) compared to the control was observed in HT-29 line treated with the extract at a concentration of 250 μg/mL. The fireweed infusion did not affect the growth of beneficial intestinal bacteria, but it did significantly inhibit *E. coli*. The cytotoxic effect of the fireweed extract indicates that it does not lose its biological activity after in vitro digestion. It can be concluded that the fireweed infusion has the potential to be used as a supporting agent in colon cancer therapy.

## 1. Introduction

*Epilobium angustifolium* (*Chamerion angustifolium*), also known as fireweed or willow herb, is a plant that belongs to the *Onagraceae* family [1]. This family includes over 200 species. Other well-known and well-described plants of this family include *Epilobium hirsutum* L. and *Epilobium parviflorum* Schreb. [2]. Fireweed is a plant with a circumpolar range. It grows in Europe, Asia, and North America. It is a medicinal plant used in traditional medicine around the world for healing numerous diseases. Externally, it is administered as an antiseptic, an anti-inflammatory agent, and a wound-healing agent. It is also used internally in the treatment of diseases of the digestive tract and the prostate [3].

The traditional use is reflected in scientific research, which shows that fireweed has antibacterial properties against a variety of gram-negative and gram-positive bacteria and therapeutic properties against benign prostatic hyperplasia (BPH). BPH is a chronic disease which affects men over 50 years of age [4]. In this condition, there is a non-cancerous enlargement of the prostate gland due to hyperproliferation of stromal and epithelial cells. It is commonly thought that the causes of BPH are associated with the androgen level, aging, and inflammation. Fireweed extracts have been shown to exhibit potent anti-BPH activity in vitro as assessed using the LNCap and BPH-1 cell lines [5].

Extracts from various species of *Epilobium* display numerous therapeutic properties, including antifungal, antimicrobial, antiproliferative, anti-inflammatory, antiandrogenic, and antioxidant effects [6,7]. Phenolic compounds, such as dolicols, triterpenoids, polyprenols, and ellagitannins, are found in extracts from these plants [8]. Alcohol extracts contain kaempferol, quercetin, myricetin, afzelin, juglalin, guajaverin, and oenothien B [9]. Oenothein B is a macrocyclic ellagitannin which shows many therapeutic properties. It reduces prostate-specific antigen (PSA) secretion in LNCaP cells, activates NK cells and T cells (giving rise to an increased expression of the activation marker), shows antibacterial activity against *Helicobacter pylori*, inhibits IL-1β and IL-6 production by activated dendritic cells, and attenuates neuroinflammation in response to LPS treatment [10]. In addition to medicines and nutraceuticals based on *E. angustifolium*, the most commonly consumed form of fireweed is a hot aqueous infusion [11], which is why we used an aqueous extract as the biological material in the present study.

There are few reports on the biological activity of phytonutrients from fireweed extracts, as tested in animal cell lines from the large intestine. This concerns both normal and cancer tissues. The only existing reports concern a dietary supplement that contains extracts of three Epilobium species. The effect of this supplement was tested on mouse colon cancer (C26 cell line). These studies confirm that these extracts can influence in vitro colorectal cell growth, cytoskeleton reorganization, and E-cadherin expression [12]. It is very important to investigate the effects of plant extracts on colon tissues, because colorectal cancer is one of the most common cancers in women and men, and its incidence continues to increase [13]. Worldwide, this cancer occurs in 614,000 cases per year in women, making it the second most common cancer in women. In contrast, it is the third most common cancer in men, causing 746,000 new cases per year. Incidence rates in less developed countries are lower by 9000 cases per year than in more developed countries. In contrast, mortality in less developed countries is higher (52% of total deaths), resulting in a low percentage of 5-year patient survival [14]. There are many risk factors for this cancer, one of which is poor diet. Digestive tract cancers are very sensitive to nutritional factors. This is particularly true of colorectal cancer, in which diet is responsible for most cases of the illness. The food people eat and the ways in which it is produced, processed, and preserved can affect human health. Therefore, diet plays a key role in cancer prevention [15] Treatment options for colorectal cancer include surgery, chemotherapy, and targeted therapy. Recently, more and more patients are turning to herbal medicines because of their therapeutic effects. In our experiments, we study plant extracts because there is much evidence that cancer cells are sensitive to substances of plant origin. Examples of such plants are *Prunella vulgaris* L., *Salvia fruticosa* L., and *Salvia oficinalis* L. Active substances, which are isolated from plants, are characterized by different mechanisms of anti-cancer activity, e.g., inhibition of nucleic acid synthesis or the clumping and destruction of cancer cells [16,17,18].

There is still scarce information in the scientific literature about the content of bioac-tive compounds in fireweed extract after intestinal digestion and the biological activity of the digested extract. In our previous work, we reported a quantitative and qualitative characterization of the phenolic compounds found in a water extract of fireweed at various stages of in vitro gastrointestinal digestion [19]. In this current study, we present quantitative determinations of selected bioactive polyphenolic compounds such as quercetine glucuronide, giaverin, afzelin, neochlorogenic acid, kaempferol gluco-side, chlorogenic acid, isoquercetin, and hyperin gallate in a fireweed extract after in vitro digestion.

The aim of our experiments was to investigate the biological activity of an *Epilobium angustifolium* infusion after in vitro digestion in a model system of intestinal epithelial tissues. Moreover, the morphological and functional status of Caco-2 cells exposed to various concentrations of the fireweed extract was analyzed using a label-free Electric Cell-Substrate Impedance Sensing assay. In vitro digestion and cell line studies provide a more precise picture of how fireweed works after ingestion, because this approach more accurately reflects the real environment of the human body. This is the first work that provides a comprehensive evaluation of fireweed extract and its effects on the human enterocyte and colon cancer cell lines.

## 2. Results

### 2.1. Chemical Characterization

The quantitative analysis was carried out for selected main compounds contained in the fireweed extract subjected to gastric and intestinal digestion. The measurements were made by comparing the measured surface areas with the surface areas of the standards. Twenty-six dominant compounds from the group of phenolic acids, flavonoids, organic acids and ester derivatives of glucosides have been identified, i.e., Citric acid, Gallic acid, Vanillic acid isomer, Galloyl-glucose, Digalloylglucose, Hydroxybenzoic acid isomer, Trigalloylglucose, Neochlorogenic acid, Hydroxybenzoic acid isomer, Coumaroyl-quinic acid isomers, Chlorogenic acid, Ferulic acid, Quercetin glucuronide, Vanillic acid isomer, Myricetin hexoside, Hyperin gallate, Myricetin glucuronide, Isoquercetin, Quercetin glucoside, Guaiaverin (quercetine arabinofuranoside), Kaempferol glucoside, Kaempferol glucuronide, Hydroxybenzoic acid isomer, Ferulic acid, Afzelin (kaempferol rhamnoside), and Caffeic acid.

The eight compounds listed in the table below were present in significant amounts. They all belonged to the group of polyphenols with a broad spectrum of anti-cancer activities (Table 1). Table 2 shows the percentage of individual compounds in the extract from *Epilobium angustifolium* L. before and after in vitro digestion.

### 2.2. Experiments on Animal Cell Lines

#### 2.2.1. MTT Assay

In this experiment, the aqueous extract tested was administered to a cancer and normal cell culture (HT-29, CCD 841 CoTr) at the following concentrations: 25 µg/mL, 50 µg/mL, 100 µg/mL, 150 µg/mL, 200 µg/mL, and 250 µg/ml. Changes in cell viability and proliferation were monitored. The results showed that the *Epilobium angustifolium* L. extract practically did not affect the viability of the cells; only at the highest concentration could statistically significant differences from the control be noticed. We can observe a 20% decrease in cell viability compared to the control. The extracts did not cause statistically significant decreases in the viability of the normal cell line after 24 h treatment, even at the highest concentrations used (Figure 1).

With regard to proliferation, statistically significant differences could be seen for both cell lines. The effect was dose-dependent. Fireweed inhibited the proliferation of the HT-29 line. This was observed even at the lowest concentration. In contrast, at a concentration of 250 μg/mL, the extract caused a decrease in proliferation to 27% compared to the control. The fireweed extract stimulated cell division of the normal cell line. Significant differences could already be seen after 96 h treatment at the concentration of 50 μg/mL, while at the highest concentration, there was an increase in proliferation up to 128% compared to the control (Figure 2).

#### 2.2.2. Neutral Red (NR) Uptake Assay

The cytotoxicity of the fireweed extract against the HT-29 and CCD 841 CoTr cell lines was investigated using the NR uptake test. Fireweed extract concentrations of 25–250 μg/mL were used. The effect of the extract was dose-dependent for both lines. Cell viability decreased in the same way in both HT-29 and CCD 841 CoTr cells. Statistically significant differences in the decrease in viability could be seen at the concentration of 150 μg/mL. The final concentration of the extract caused a decrease in the number of viable cells to 50% in the cancer line and to 58% in the normal line (Figure 3).

#### 2.2.3. May–Grünwald–Giemsa Staining

The staining experiment was performed to confirm the results of the spectrophotometric test (MTT assay). The cancer cell line HT-29 and the normal cell line CCD 841 CoTr as well as the fireweed extract at concentrations of 100 and 250 μg/mL were used. Cancer cell morphology changed after incubation with the extract (250 μg/mL) compared to the control. In the control and at the fireweed extract concentration of 100 μg/mL, a large number of clustered cells were observed; the cytoplasm and cell nuclei were very clearly visible. When a concentration of 250 μg/mL was used, much fewer cells were visible, and they did not tend to clump together. It was very difficult to distinguish between the cell nucleus and the cytoplasm because the cells changed their shape. Normal cell morphology looked similar in the control and after incubation with the extract. Spindle-shaped cells were clearly discernible. One could easily discriminate between the cytoplasm and the cell nuclei. At a higher concentration, the number of cells decreased (Figure 4).

#### 2.2.4. Electric Cell-Substrate Impedance Sensing (ECIS)

The ECIS system is used to detect and quantify morphological changes. When cells are stimulated with test substances, changes occur in their metabolism and functionality, which directly lead to changes in the impedance, resistance, and capacitance recorded by the device. In the present experiment, the in vitro digested fireweed extract was administered to the CaCo-2 cell culture run in the ECIS system at the following concentrations: 25 µg/mL, 50 µg/mL, 100 µg/mL, 150 µg/mL, 200 µg/mL, and 250 µg/mL. Changes in impedance, resistance, and capacitance in the cell culture were monitored over a period of 200 h. The growth medium was not changed throughout the time of the experiment, and the environment and the measurement conditions were kept constant. The graphs show the concentrations with the largest differences (Figure 5).

In the case of the initial concentrations up to 100 µg/mL, differences in the impedance values were found between the control cell culture and cells treated with the fireweed extract. Over the first 24 h of cultivation, before the in vitro digested fireweed extract was added to the test culture, the impedance values for both cultivation variants remained relatively similar. After 60 h into the measurement, a significant difference in impedance was noted: about 800 Ohms for cells in the control culture and 500 Ohms for the test culture. Up to 140 h, the growth of both cultures stabilized, and only after 150 h did the impedance in the control culture increase to 1000 Ohm, while in the culture with the addition of the test substance, it remained relatively stable at about 700 Ohm. The data obtained lead to the conclusion that the capacitance in the control cultures decreased slightly more and reached the lowest value of about 5–10 nF between 40 and 200 h. A clear downward trend in impedance and resistance values could be observed with the increasing fireweed extract concentration. This may indicate that as the concentration of the fireweed extract increased, the cells were able to absorb larger amounts of it. In other words, the higher the concentration of the in vitro digested fireweed extract, the greater was its absorbability and the clearer was the manifestation of changes in the electrical parameters.

#### 2.2.5. Antimicrobial Properties

The optical density experiment was carried out using six strains of Gram-negative and Gram-positive bacteria: Lactobacillus rhamnosus, Enterococcus faecalis, Enterobacter cloacae, Escherichia coli, Bifidobacterium adolescentis, and Bifidobacterium longum. The in vitro digested fireweed extract was added to the cultures at the concentrations of 16, 32, 64, 128, and 250 μg/mL. It can be seen from the results that the fireweed extract did not cause changes in the growth of L. rhamnosus, E. cloaceae, B.longum, and B.adolescentis; their OD_600_ did not change under the influence of the extract and was practically the same as in the control. The fireweed extract stimulated the growth of E. faecalis. The OD_600_ for this bacterium at the highest fireweed extract concentration was twice as high as in the control, while the doubling time of the generation was almost twice as low. On the other hand, in the case of E. coli, bacterial growth was inhibited by fireweed extract and its generation time was almost doubled compared to the control (Table 3). After analyzing the results obtained, it can be concluded that the best antimicrobial concentration is a concentration of 64 μg/mL, because maxOD_600_ of E. coli is twice as low compared to the control. In contrast, the growth stimulation of E. faecalis is still insignificant.

## 3. Discussion

Our previous work shows that fireweed infusion is rich in phenolic compounds, with flavonoids being the most abundant at 53.04 ± 1.24 mg quercetin/g of the freeze-dried infusion. The HPLC-MS analysis confirmed the presence of phenolic acids (e.g., gallic, vanillic, hydroxybenzoic, ferulic, and chlorogenic acids), flavonoid glucosides (e.g., quercetin, kaempferol, and myricetin derivatives), and galloyl glucose derivatives. However, we did not ascertain the presence of oenothein B, which is unique to fireweed, which could have resulted from the fact that we used water for the infusion instead of organic solvents [19]. Other authors also reported that fireweed is rich in polyphenols. Kosalec et al. (2008) Ref [4] showed that ethanol extracts from fireweed flowers and leaves were rich in phenol derivatives—10.04 ± 0.35% and 12.96 ± 0.32% of dry weight, respectively. Previously, in various *E. angustifolium* extracts, certain compounds have been detected, such as ellagitannin, oenothein B, well-known and widely distributed polyphenols such as kaempferol, quercetin, myricetin, and their derivatives, phenolic acids (ellagic, chlorogenic, p-coumaric, gallic, protocatechuic, gallic, cinnamic, caffeic, and ferulic acids), valoneic acid dilactone, terpene compounds, and other substances [20,21]. It is these compounds that are responsible for the beneficial effects of E. angustifolium, including its antioxidant, anti-inflammatory, inhibitory enzyme, antitumor, antimicrobial, and immunomodulatory activities, as confirmed in in vitro and in vivo studies [5].

In this work, we were especially interested in the quantification of the bioactive compounds present in a fireweed infusion after in vitro digestion. We identified quercetine glucuronide, guaiaverin, and afzelin as quantitatively dominant compounds, and zeochlorogenic acid, kaempferol glucoside, chlorogenic acid, isoquercetin, and hyperin gallateother as compounds present in smaller quantities (at a similar level). Comparing these results with our previous study, in which we had determined the content of phenolic compounds in a fireweed infusion (prepared with the same method) [19], we found that all the polyphenolic compounds we identified in the present study had also occurred in *E. angustifolium* water extract before in vitro digestion. However, we observed a decrease in the total polyphenolic content during the small intestinal digestion phase [19]. Similarly, Dacrema et al. [21] reported a drop in the content of individual polyphenolic compounds after in vitro digestion of fireweed extract. They observed a loss of individual polyphenolic compounds after the orogastric phase in the range of 1.92–84.17%, and a 11.83–98.07% decrease after the duodenal phase. The loss of polyphenolic compounds during digestion seems indisputable, but it is important to point out that these authors also identified quercetine, kaempferol, and chlorogenic acid in fireweed extract after in vitro digestion. Some differences in the results of our study and Dacrema et al.’s work may be due to differences in the in vitro digestion procedures used.

Other authors who investigated extracts of *Epilobium* plants (*E. rosmarinifolium, E. spicatum*, and *E. tetragonum*) have observed growth-inhibitory effects in different human cell lines (PZ-HPV-7, normal prostate cells; LNCaP, transformed prostate cells; HMEC, mammary cells, and 1321N1, astrocytoma cells). They revealed that the kaempferol and quercetin present in *Epilobium* plants showed antiproliferative effects [22]. *Epilobium* species have well-described antioxidant and anti-inflammatory properties, which may enhance their anti-cancer effects [23]. Thus, in our experiment, the inhibition of the growth of the intestinal cancer cells HT-29 can be attributed to the activity of quercetin and kaempferol, which were the most abundant phenolic compounds in the fireweed extract after in vitro digestion. *Epilobium angustifolium* is well known for its therapeutic effects on prostate diseases and breast cancer [23,24]. However, to the best of our knowledge, its potential anti-proliferative and cytotoxic effects on human colon cancer cells have not been studied yet.

Plant extracts are complex matrices of compounds that tend to interfere with the label-dependent methods that are typically used for cytotoxicity screening, also the ones employed in this work. Therefore, we additionally applied a label-free ECIS-based cytotoxicity assay for the assessment of the cytotoxicity of the in vitro digested fireweed extract.

The impedance curves showed an overall concentration- and time-dependent toxic effect of the fireweed extract. As expected, lower concentrations (25 and 100 µg/mL) were less cytotoxic, while the higher concentrations (200 and 250 µg/mL) caused more acute effects, with impedance, measured during the first 60 h of the experiment, decreasing to around 50% compared to untreated cells. The cytotoxicity detected in the fireweed extract can be explained by the chemical composition of the infusion, which does not lose its biological activity after the in vitro digestion process. These observations are in line with the results we obtained using other, label-dependent cytotoxicity screening methods. There was a clear dose-dependency at the concentrations of 200 and 250 µg/mL. As the concentration increased, the impedance and resistance values fell. This may indicate that, at the higher concentrations, the cells absorbed more of the fireweed extract. Changes in electrical parameters reflect alterations in the morphological and functional state of the cell, which can also be observed after May-Grünwald-Giemsa staining. The physiological processes taking place in cells are mirrored by changes in morphology and, thus, also the effects of absorption of bioactive substances added to the culture. It can be concluded that the higher the concentration of the in vitro digested fireweed extract, the greater the absorbability and the clearer the manifestation of changes in electrical parameters. Our observations are confirmed by other studies which showed that extracts from *E. angustifolium* exerted an inhibitory effect on the proliferation of human cancer cell lines and inhibited DNA synthesis in human astrocytoma cells 1321N1 [25]. Moreover, an aqueous extract of *E. angustifolium* demonstrated a higher antiproliferative activity than ethanol extracts [26].

To the best of our knowledge, the influence of *Epilobium* plant extracts on the growth of lactic acid bacteria and human gut microbiome has not been studied yet. Our results clearly show that fireweed infusion does not adversely affect the growth of beneficial intestinal bacteria. However, a commensal strain of *E. coli* was significantly inhibited. This is consistent with the results of studies by other authors [27]. According to Bartfay et al. [3], *E. angustifolium* extracts display a higher antibacterial activity against *S. aureus*, *E. coli*, and *P. aeruginosa* than antibiotics. This is a very important observation in the light of the possibility of using fireweed infusion as a supporting agent in colon cancer therapy. The beneficial gut microbiota plays an essential role in regulating intestinal homeostasis. On the other hand, microbial dysbiosis and the emergence of colorectal-cancer-associated microorganisms, such as *E. coli*, can lead to intestinal disease and exert potentially carcinogenic effects on the host [28]. Taxonomic (16S rRNA gene-based) and metagenomic (pan-genomic-based) microbial sequence analyses have provided a clear link between bacteria, inflammation, and colorectal cancer [29,30]. Evidence from functional studies indicates that *E. coli* utilize a complex arsenal of virulence factors to colonize and persist in the intestine. Some of these virulence factors, such as genotoxin and colibactin, have been found to promote colorectal cancer in experimental models [28].

## 4. Materials and Methods

### 4.1. Materials

#### Plant Material

We obtained the plant material for research from the herbal company Polskie Zioła, Piaski Wielkie, Poland. Extraction was performed according to Kiss et al. [31] with slight modifications. A thirty-minute extraction with distilled water at a temperature of 90 °C was performed with the use of ultrasound for 10 min (1:10 plant-to-solvent ratio (*m/v*)). Then, the infusions were filtered and after cooling, subjected to in vitro digestion. Digested samples were centrifuged (10,700× *g* for 30 min at 4 °C) and frozen at 80 °C for 24 h. Next, lyophilization was carried out in a Labconco freeze dryer (Labconco, Kansas City, MO, USA).

### 4.2. Qualitative and Quantitative Analysis of Samples by HPLC-ESI-QTOF-MS/MS

This method was performed as described in our previous work [32].

### 4.3. In Vitro Gastrointestinal Digestion of Fireweed Infusion

In vitro digestion was performed as described in our previous work [32].

### 4.4. Cell Cultures

The human colon adenocarcinoma cell line HT-29 and the human colon epithelial cell line CCD 841 CoTr were purchased from the American Type Culture Collection (ATCC, Manassas, VA, USA). HT-29 cells were grown in RPMI 1640 medium with 10% or 2% FBS. CCD841 CoTr cells were cultured in mixed media: RPMI 1640 and Dulbecco (DMEM) (1:1), with a 10% or a 2% addition of FBS. All media were supplemented with penicillin (100 U/mL) and streptomycin (100 mg/mL). The cells were maintained in standard conditions: temperature 37 °C (HT-29), 34 °C (CCD 841 CoTr), and 5% CO_2_. CaCo-2 cells for Electric Cell-Substrate Impedance Sensing (ECIS) were purchased from ATCC (ATCC HTB-37™, Manassas, VA, USA) and cultured according to the manufacturer’s instructions. The cells were maintained in Dulbecco’s Modified Eagle’s Medium supplemented with 20% FBS Good HI (fetal bovine serum) and antibiotics—100 IU/mL Streptomycin/Penicillin/Amphotericin B. The process was carried out in a Galaxy 170R incubator under controlled growth conditions, constant humidity, and 5% CO_2_ air saturation.

### 4.5. MTT Cytotoxicity Assay

The MTT assay was used to determine cell viability and proliferation. In the assay, the water-soluble yellow dye MTT [3-(4,5-dimethylthiazol-2-yl)-2,5-diphenyltetrazolium bromide] is converted to insoluble purple formazan by the action of mitochondrial reductase [33]. The cells were seeded on 96-well microplates at a density of 1 × 10^5^ cells/mL. The next day, the culture medium was removed, and the appropriate concentrations of the in vitro digested fireweed extract (25, 50, 100, 150, 200, and 250 µg/mL) in fresh medium were added to the cells. After a 24-h incubation with or without the examined extract, MTT solution (5 mg/mL in PBS) was added to the cells for 3 h. Next, SDS solution (10% SDS in 0.01N HCl) was added to all wells to stop the reaction and dissolve the resulting formazan crystals. After 24 h of incubation, the absorbance at λ = 570 nm was determined spectrophotometrically using a Varioskan™ LUX multimode microplate reader (Thermo Fisher Scientific, Waltham, MA, USA). The amount of formazan formed (color intensity), read spectrophotometrically, was directly proportional to the number of live, undamaged cells.

### 4.6. MTT Proliferation Assay

The MTT assay was used to determine cell proliferation. The experiment was carried out exactly as described above. The only two differences were the number of cells used (100 µL of 5 × 10^4^ cells/mL) and the time of incubation of the cells with the in vitro digested fireweed extract (96 h).

### 4.7. Neutral Red (NR) Uptake Assay

This method was performed as described in work by Ates et al. [34]. The NR uptake method was used to determine cell viability. The test is based on the accumulation of NR in living lysosome cells. This process takes place by means of passive transport. Dead and damaged cells do not bind the dye. The cells were seeded on 96-well microplates at a density of 1 × 10^5^ cells/mL. The next day, the culture medium was removed and the appropriate concentrations of the in vitro digested fireweed extract (25, 50, 100, 150, 200, and 250 µg/mL) in fresh medium were added to the cells. After a 24-h incubation, the culture media were removed, and NR (40 µg/mL) was added to the cells for 3 h at 37 °C. Next, all wells were rinsed with formalin (0.5% in 1% CaCl_2_). After 2 min, extraction was performed for 20 min with shaking at room temperature, adding glacial acetic acid (1% in 50% ethanol). Thereafter, the absorbance at λ = 550 nm was determined spectrophotometrically using the Varioskan™ LUX multimode microplate reader (Thermo Fisher Scientific, Waltham, MA, USA). The amount of the dye absorbed (color intensity), read spectrophotometrically, was directly proportional to the number of living, undamaged cells.

### 4.8. May-Grünwald-Giemsa Staining

In the May-Grünwald-Giemsa reaction, cells are stained violet-blue. This method was used to visualize the number and morphology of culture cells and how these parameters changed under the influence of the test extract. The staining was performed on 24-well plates (1 mL 1 × 10^5^ cells/mL). After a 24 h incubation, the culture medium was removed, and appropriate concentrations of the in vitro digested fireweed extract (25, 50, 100, 150, 200, and 250 µg/mL) in fresh medium were added to the cells. The next day, the culture media were removed, and the cells were stained with the May-Grünwald dye for 3 min. Then, deionized water was added, after which the liquid was removed. In the next step, the cells were stained with the Giemsa dye (dilution 1:20) for 30 min at room temperature. After this time, the dye was removed, and the wells were rinsed with deionized water and allowed to dry. Images were taken using the Motic^®^ AE31 Elite inverted microscope (Motic Scientific, Richmond, BC, Canada).

### 4.9. Electric Cell-Substrate Impedance Sensing (ECIS)

An ECIS system Zθ instrument (Applied Biophysics Ltd., Troy, NJ, USA) was used to measure impedance, resistance, and capacitance. The system consists of two separate units: a docking station with two eight-well culture matrices, located inside the incubator, and a Z theta controller unit, located outside the incubator. For the experiment, 8W10E electrodes were used, which comprised eight wells and 10 active electrodes in each well. The matrix station is responsible for the additional signal processing and switching mechanism to change the measurement from one well to the next well. The device measures both resistance and capacitance over a wide range of alternating current frequencies. Moreover, the device only accepts signals of the same frequency as the source oscillator frequency and also eliminates most of the external electrical noise. The ECIS measuring electrodes were placed in the docking station in an incubator (Galaxy 170R) under controlled conditions of temperature (37 °C) and carbonation (5% CO_2_). The culture plates were then incubated for 24 h with the culture medium (Dulbecco’s Modified Eagle’s Medium) overnight. After stabilization, the matrix was removed from the station and inoculated with cells. Matrix inoculation was performed using 600 microliters per well of the cell suspension of ~1.2 × 10^5^ cells/mL. The in vitro digested fireweed extract was added to the inoculated wells at final concentrations of 25 µg/mL, 50 µg/mL, 100 µg/mL, 150 µg/mL, 200 µg/mL, and 250 µg/mL. After the cell culture had been supplemented with the test preparation, the culture plates were returned to the incubator and real-time measurements were started. The maximum response for Z, R, and C occurs at different frequencies by default. In our experiment, we used the default optimal frequencies: resistance (R) 4000 Hz, impedance (Z) 32,000 Hz, and capacitance (C) 64,000 Hz. Changes were recorded as impedance signals and the resulting data were processed using the ECIS software. The data generated were impedance/resistance/capacitance versus time.

### 4.10. Bacterial Strains and Culture Method

The bacterial species used in this study were selected to represent a diverse group of intestinal bacteria. Both Gram-positive and Gram-negative bacteria were selected, as well as microorganisms classified as beneficial and potentially pathogenic microflora. *Lactobacillus rhamnosus* B-445, *Enterococcus faecalis* JCM 1513, *Escherichia coli*, *Bifidobacterium adolescentis* DSM 20083, *Bifidobacterium longum* DSM 20088, and *Enterobacter cloacae* PCM 533 were used in this study. *L. rhamnosus* was cultured in MRS medium (OXOID LTD., Hampshire, UK), *E. faecalis* and *E. cloacae* in nutrient broth (BTL, Łódź, Poland), *E. coli* in LB broth (BTL, Łódź, Poland), and the *Bifidobacterium* strains were cultivated in modified Garche’s medium containing peptone, 20 g/L; lactose, 10 g/L; sodium acetate, 6 g/L; Na_2_HPO_4_ × 12 H_2_O, 2.5 g/L; yeast extract, 2 g/L; KH_2_PO_4_, 2 g/L; L-cysteine hydrochloride, 0.4 g/L; MgSO_4_ × 7H_2_O, 0.12 g/L; pH 6.4. All the test strains were incubated for twenty-four hours at 37 °C.

#### Effect on Bacterial Growth

The effect of the in vitro digested fireweed extract on the growth of the selected bacterial strains was determined by measuring the optical density (OD_600_) in microcultures with a Bioscreen C system (Labsystem, Helsinki, Finland). Bacterial cultures were centrifuged and diluted in physiological saline to an optical density of 0.2 at 600 nm. Subsequently, 50 μL of the bacterial suspension was added into each well of a sterile 100-well Bioscreen plate containing 250 μL of the appropriate culture medium with the extract concentrations tested (16, 32, 64, 128, 250 μg/mL, respectively). Each concentration was tested in triplicate for each bacterium. Three wells containing a bacterial suspension with no fireweed extract (compound-free growth control) and three wells containing growth media alone (background control) were included in this plate. OD_600_ was measured for 48 h at 37 °C at 600 nm every 2 h for each well. To better determine the effect of fireweed extracts on individual gut bacteria growth curve parameters were determined using the PYTHON script according to Hoeflinger et al. [35]. Parameters such as lag time or doubling time, which determine changes in the time of adaptation of particular bacteria to environmental conditions and the length of generation time, were determined.

### 4.11. Statiscical Analysis

The results of the experiments were collated in MS Excel 2013 (Microsoft Corporation, Redmont, WA, USA). Statistical data were analyzed with a one-way ANOVA test and post-hoc Dunnett’s test (all columns compared to the control) using STATISTICA 13.3 (StatSoft, Cracow, Poland). This study used *p*-values to describe statistically significant data; * *p* < 0.01, ** *p* < 0.005, *** *p* < 0.001 were considered significant compared to control.

## 5. Conclusions

Our results revealed that *E. angustifolium* infusion still retained its biological properties after in vitro digestion. Despite the fact that the total amount of phenolic compounds in the fireweed infusion decreased after in vitro digestion, polyphenols such as quercetin or kaempferol, which are responsible for the cytotoxic effect of the extract on cancerous colon cells, still remained in it at a sufficiently high level. The fireweed infusion adversely affected the morphology and viability of colon cancer cells and inhibited the proliferation of the HT-29 line. The cytotoxic effect of the in vitro digested fireweed extract was dose- and time-dependent. Higher concentrations (200 and 250 µg/mL) caused the most acute cytotoxic effects. The fireweed infusion did not inhibit the growth of beneficial intestinal bacteria of the genera *Lactobacillus* or *Bifidobacterium*. However, a commensal strain of *E. coli* was significantly inhibited. Our work is the first to comprehensively investigate an in vitro digested fireweed extract and its cytotoxic effects on the human colon cancer cell lines. The results we obtained show that a fireweed infusion can be used as a factor supporting the therapy of colon cancer; however, the extract must be checked in normal cells and in animal models in order to evaluate its toxicity. Further investigations should be carried out on *E. angustifolium* extract-based food supplements, the results of which might support the use of *E. angustifolium* extract in the early stages of colorectal cancer.

## Figures and Tables

**Figure 1 molecules-27-01006-f001:**
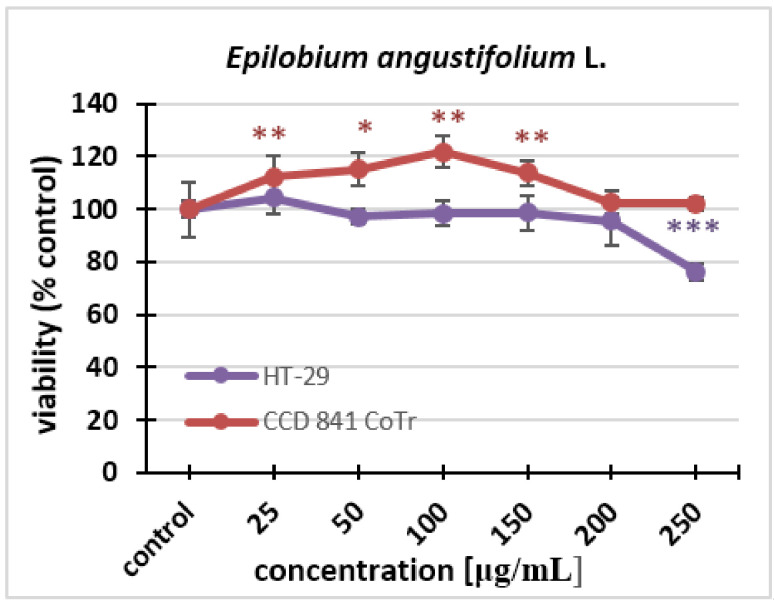
The effect of the fireweed extract on the viability of HT-29 and CCD 841 CoTr cells analyzed after 24 h using the MTT assay. The values are compared to the control (100% viability); * *p* < 0.01, ** *p* < 0.005, *** *p* < 0.001, one-way ANOVA, Dunnett’s test.

**Figure 2 molecules-27-01006-f002:**
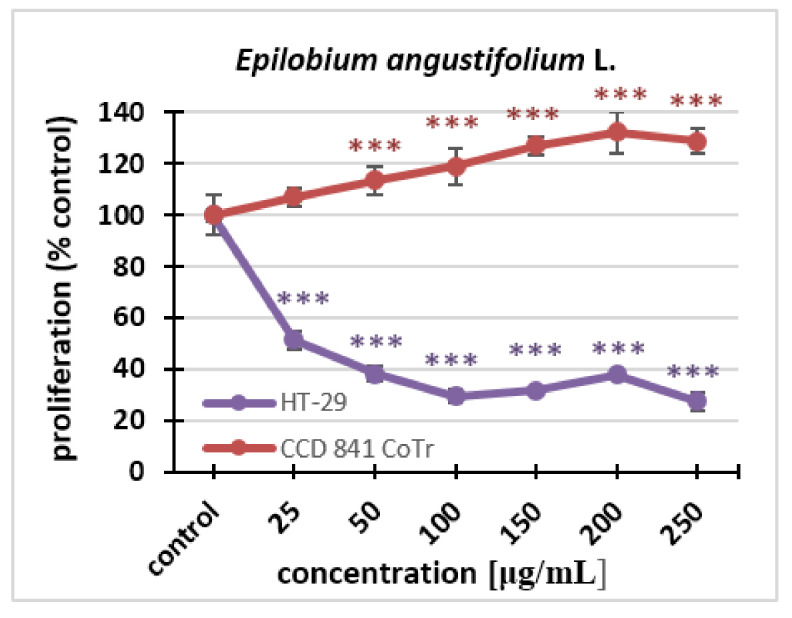
The influence of the *Epilobium angustifolium* L. extract on the proliferation of HT-29 and CCD 841 CoTr cells analyzed after 96 h using the MTT assay. The values are compared to the control (100% proliferation); *** *p* < 0.001, one-way ANOVA, Dunnett’s test.

**Figure 3 molecules-27-01006-f003:**
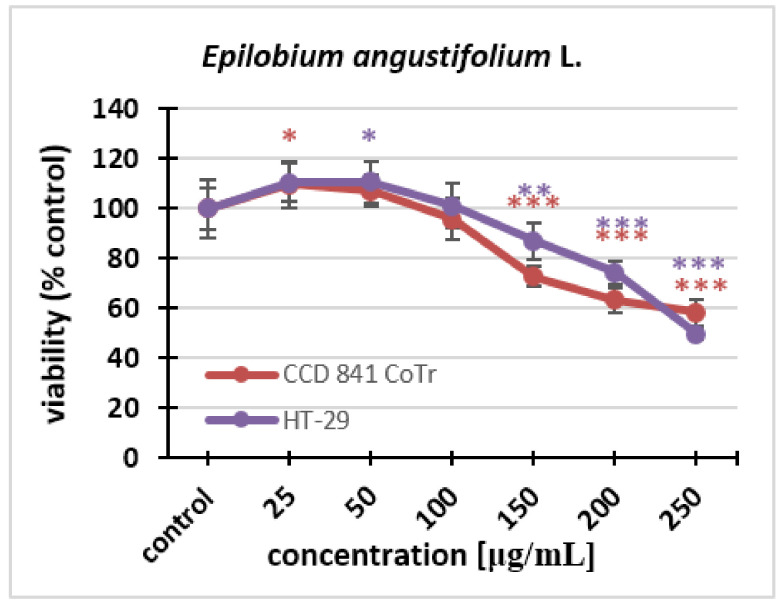
Cytotoxicity of the fireweed extract on HT-29 and CCD 841 CoTr cell lines was determined using the neutral red (NR) uptake assay. The values are compared to the control (100% viability); * *p* < 0.01, ** *p* < 0.005, *** *p* < 0.001, one-way ANOVA, Dunnett’s test.

**Figure 4 molecules-27-01006-f004:**
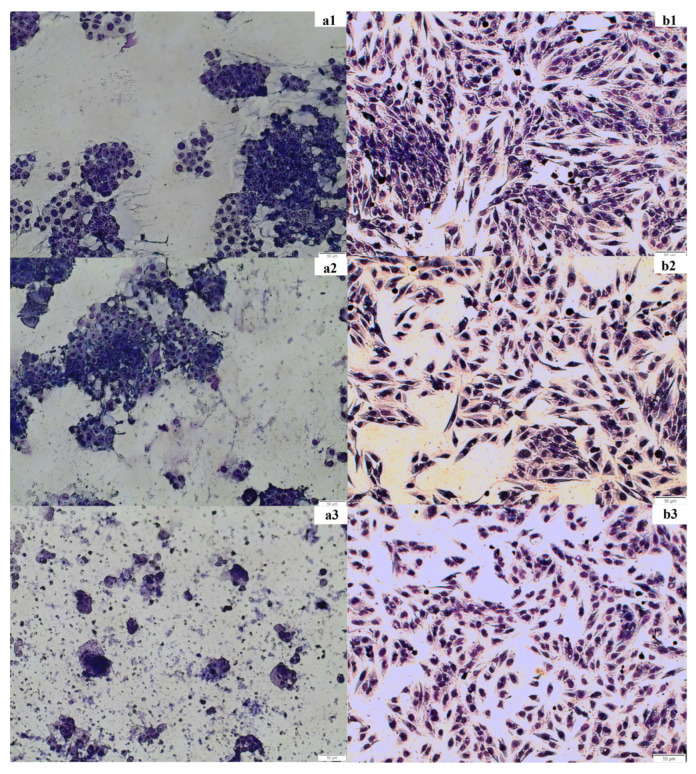
Changes in the morphology and number of HT-29 and CCD 841 CoTr cells under the influence of fireweed extract. Images were taken using a MOTIC AE31E series light microscope, magnification 100× *g*. (**a1**)—control, HT-29; (**a2**)—100 μg/mL, HT-29; (**a3**)—250 μg/mL, HT-29 and (**b1**)— control, CCD 841 CoTr; (**b2**)—100 μg/mL, CCD 841 CoTr; (**b3**)—250 μg/mL, CCD 841 CoTr.

**Figure 5 molecules-27-01006-f005:**
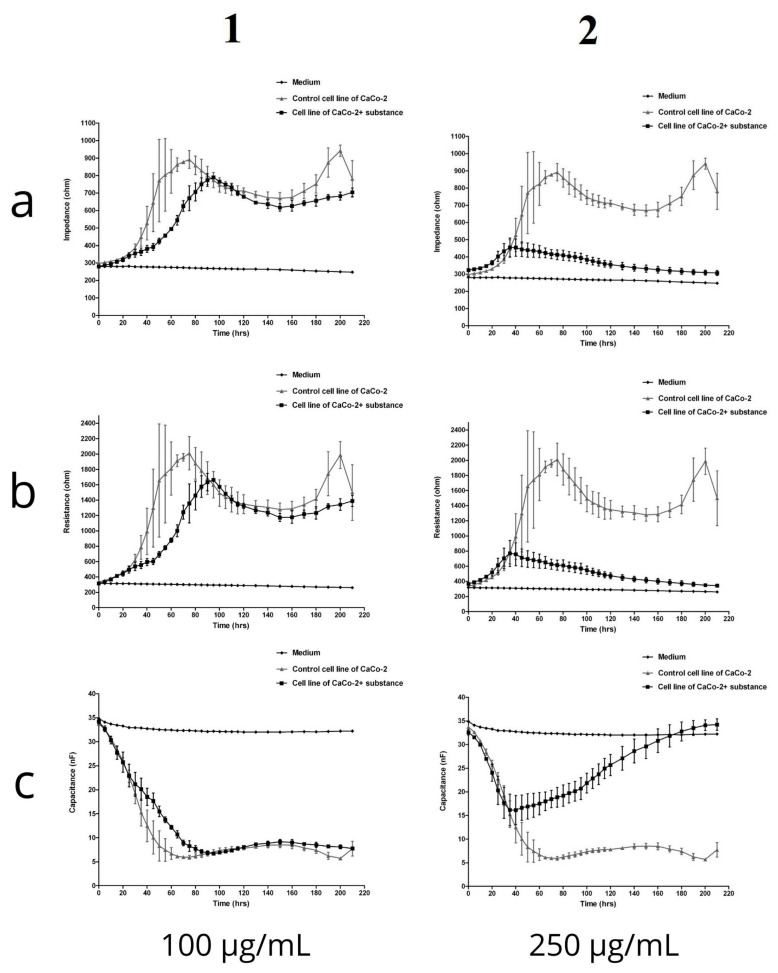
Changes in the impedance, resistance, and capacitance of CaCo-2 cells after incubation with the in vitro digested fireweed extract: (**a1**)—impedance, (**b1**)—resistance, and (**c1**)—capaticance, concentration: 100 μg/mL; (**a2**)—impedance, (**b2**)—resistance, and (**c2**)—capaticance, concentration: 250 μg/mL. The experiment was conducted in the ECIS system.

**Table 1 molecules-27-01006-t001:** The content of the individual compounds (in μg/mL) in the extract from in vitro digested *Epilobium angustifolium* L. The determination was performed using a high-resolution mass spectrometer coupled with a high-performance liquid chromatograph (HPLC-ESI-Q-TOF-MS).

Name of Substance	Content in μg/mL after Digestion
Quercetine glucuronide	593.0
Giaverin	212.7
Afzelin	148.2
Neochlorogenic acid	37.3
Kaempferol glucoside	24.9
Chlorogenic acid	20.8
Isoquercetin	15.6
Hyperin gallate	15.5

**Table 2 molecules-27-01006-t002:** The percentage of individual compounds in the extract from *Epilobium angustifolium* L. before and after in vitro digestion. The determination was performed using a high-resolution mass spectrometer coupled with a high-performance liquid chromatograph (HPLC-ESI-Q-TOF-MS). SD —standard deviation and RSD—relative standard deviation (in percent) of the results.

Name of Substance	Before Digestion	SD	RSD	After Digestion	SD	RSD
Neochlorogenic acid	0.28796	0.00434	1.5	0.00370	0.00012	3.4
Chlorogenic acid	0.09889	0.00279	2.8	0.00239	0.00009	3.8
Isoquercetin	0.44318	0.01478	3.3	0.00149	0.00013	8.8
Kaempferol glucoside	0.59884	0.06533	10.9	0.00252	0.00007	2.8
Hyperin gallate	1.48623	0.02287	1.5	0.00236	0.00008	3.6
Quercetine glucuronide	4.75316	0.09163	1.9	0.06274	0.00058	0.9
Giaverin	0.71688	0.01206	1.7	0.02349	0.00004	0.2
Afzelin	1.11672	0.03015	2.7	0.02849	0.00062	2.2

**Table 3 molecules-27-01006-t003:** Antimicrobial effect of in vitro digested extract of *Epilobium angustifolium* L. Experiment performed using the Bioscreen C reader. The results were analyzed in Python 3.9.7.

Strain	Concentration of Fireweed Extract [μg/mL]	Lag Time (h)	Doubling Time (h)	Min OD	Max OD	Effect on Bacterial Growth
*Lactobacillus rhamnosus*	Control *	12.12	3.24	0.43	1.99	
	16	14.97	3.58	0.38	1.91	no differences in growth
	32	14.78	3.50	0.40	1.94	
	64	14.86	3.51	0.40	1.95	
	128	13.96	3.88	0.39	1.95	
	250	13.14	3.75	0.41	1.98	
*Enterococcus faecalis*	Control *	7.46	10.20	0.19	0.61	
	16	7.26	10.74	0.19	0.60	
	32	6.47	10.38	0.18	0.76	stimulation of growth
	64	6.47	11.75	0.18	0.77	
	128	5.30	8.97	0.19	1.07	
	250	7.28	6.18	0.18	1.35	
*Enterobacter cloacae*	Control *	0.52	5.47	0.19	1.29	
	16	0	6.25	0.27	1.20	
	32	0	5.92	0.19	1.17	no differences in growth
	64	0	5,67	0.14	1.17	
	128	0	6.42	0.19	1.20	
	250	0	6.05	0.22	1.32	
*Escherichia coli*	Control *	0	14.85	0.20	1.41	
	16	0	16.10	0.22	1.05	
	32	0	15.96	0.24	1.07	MIC-64 μg/mL
	64	0	20.21	0.22	0.84	
	128	0	16.74	0.19	0.83	
	250	0	25.12	0.25	0.87	
*Bifidobacterium adolescentis*	Control *	4.53	33.25	0.30	0.78	
	16	6.98	38.41	0.28	0.67	
	32	2.77	40.73	0.24	0.69	no differences in growth
	64	9.15	42.12	0.25	0.70	
	128	3.77	39.43	0.28	0.74	
	250	0	93.21	0.39	0.69	
*Bifidobacterium longum*	Control *	4.66	34.13	0.27	0.75	
	16	5.72	46.14	0.25	0.64	
	32	8	52.48	0.26	0.63	no differences in growth
	64	6.72	41.14	0.25	0.63	
	128	0	48.85	0.26	0.61	
	250	0	73.18	0.27	0.53	

* Control—bacterial growth control without test extract.

## Data Availability

Not applicable.

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
