# Peer review of "Biological Activity of an Epilobium angustifolium L. (Fireweed) Infusion after In Vitro Digestion"

_molecules, 2022, doi:10.3390/molecules27031006_

Round 1
Reviewer 1 Report
The authors presented the anticancer and antibacterial activity of invitro digested extract of Epilobium angustifolium L. in colon cancer cell line and different types of bacteria. The results presented in this MS is the continuation of their previous published work in which they already have presented the phytochemicals difference present in undigested and in vitro digested extract.
The first thing that the authors must have to do is to get the English language edited by native speaker, or professional editing service.
Table-2; SD and RSD should be shown in column next to the compound not in separate row
Figure-2: This is very confusing, the same extract shown not active in figure 1 and shown highly active in figure 2 by same assay. Is it same type of extract? Is it that the authors wanted to show the effect of extract on cell lines before digestion and after digestion ? Please clear it
Table-3 a) Why undigested extract was not used as control in this experiment?
b) Calculate MIC (minimum inhibitory concentration) instead of showing min OD and max OD for each microorganism.
The most important factor is the evaluation of the toxicity of the invitro digested extract in animal system, in order to determine the therapeutic application of the extract after digestion, it should not be toxic in animals.
In text the authors should word substance instead of extract which should be avoided.
Author Response
Thank you for the thorough review, which helped us to improve the publication. Authors have considered all Reviewers’ comments and corrected manuscript according to major and minor concerns.
Reviewer 1
The authors presented the anticancer and antibacterial activity of in vitro digested extract of Epilobium angustifolium L. in colon cancer cell line and different types of bacteria. The results presented in this MS is the continuation of their previous published work in which they already have presented the phytochemicals difference present in undigested and in vitro digested extract.
The first thing that the authors must have to do is to get the English language edited by native speaker, or professional editing service.
The English language has been edited by a native speaker.
Table-2; SD and RSD should be shown in column next to the compound not in separate row
We agree with the Reviewer, table have been changed according to the comments.
Figure-2: This is very confusing, the same extract shown not active in figure 1 and shown highly active in figure 2 by same assay. Is it same type of extract? Is it that the authors wanted to show the effect of extract on cell lines before digestion and after digestion ? Please clear it
Figures show the effect of fireweed extract on normal and cancer cell lines. Figure 1 shows the effect of fireweed extract on the viability of HT-29 and CCD 841 CoTr cells, incubated for 24 h, and figure 2 shows the effect of Epilobium angustifolium L. extract on the proliferation of HT-29 and CCD 841 CoTr cells, incubated with the extract for 96 h. In both cases, the extract was after in vitro digestion. The MTT method was used here, which can be applied both to determine cell proliferation and viability when testing substances with cytotoxic and antiproliferative activity. It involves measuring mitochondrial oxidation-reduction activity using the MTT dye (3- (4,5-dimethylthiazil-2-yl) -2,5-diphenyltetrazolium bromide), which in living cells is reduced to purple formazan crystals by succinate dehydrogenase. In toxicological studies, besides the assessment of the doses of the analyzed substances, the time of exposure of cells to the tested compound is equally important. It has been shown that short exposure time is mainly associated with impaired cell function. Longer exposure affects proliferation and cell cycle changes.
Table-3 a) Why undigested extract was not used as control in this experiment?
The aim of our microbiological analyzes was to check how the fireweed infusion influences the growth of intestinal microbiota (bacteria cells). Therefore, we added different concentrations of fireweed extract after in vitro digestion to the bacterial culture. We did not include the extract before digestion in the research, because the intestinal microbiota does not come into contact with such an extract under natural conditions.
- b) Calculate MIC (minimum inhibitory concentration) instead of showing min OD and max OD for each microorganism.
In our study, we did not calculate MIC because the fireweed extract did not cause changes in the growth of L. rhamnosus, E. cloaceae, B.longum and B.adolescentis. Their OD600 did not change under the influence of the extract and was virtually the same as in the control. Fireweed extract stimulated the growth of E. faecalis. The OD600 value for this bacterium at the highest concentration of the extract was twice that of the control. The extract inhibited the growth of only E. coli, the growth of the bacteria was inhibited by the fireweed extract and the generation time was almost twice as long compared to the control. Information on bacterial growth was included in the table as noted. The MIC was determined for E. coli.
The most important factor is the evaluation of the toxicity of the invitro digested extract in animal system, in order to determine the therapeutic application of the extract after digestion, it should not be toxic in animals.
We agree with the reviewer, the following statement occurs in the conclusion: "Our results indicate that the extract of fireweed can be used as an adjuvant therapy for colorectal cancer, but in vivo or human tissue studies need to be performed to fully confirm this observation."
In text the authors should word substance instead of extract which should be avoided.
In our opinion, the use of the word extract is preferable, because the extract is considered to be a solution of chemical compounds that has been extracted by an extraction process. This solution can be either one particular substance or a mixture of several different components. It can also be a finished product on its own, but it can also be subjected to other chemical processes in order to obtain the purest possible form of the desired substance, e.g. an active substance, i.e. a chemical compound whose action produces a therapeutic effect.

Reviewer 2 Report
Dear Authors, In this manuscript, Polak-Berecka and coworkers described biological activity of an Epilobium angustifolium L. infusion after in vitro digestiona. The individual compounds in the extract inhibited the proliferation of the HT-29 line. Antimicrobial properties were also investigated in the growth of L. rhamnosus, E. cloaceae, B. longum and B. adolescentis. However, this work did not show outstanding activities.
1.Line 18, human colon epithelial cell line CCD 841 841 shoule be CCD 841. The same as in line 349.
2.Line 26, double full stop.
3.Line 52, one more space in effects [6,7].
4.Line 107, following 2. Results, it should be 3.1.
5.Line 168, one more space 250 ug/mL.
6.Line 196, double Figure 5.
7.Lines 354, 360, and following, correct CO2.
8.Line 385, correct 1x105.
Author Response
Thank you for the thorough review, which helped us to improve the publication. Authors have considered all Reviewers’ comments and corrected manuscript according to major and minor concerns.
Reviewer 2
Dear Authors, In this manuscript, Polak-Berecka and coworkers described biological activity of an Epilobium angustifolium L. infusion after in vitro digestiona. The individual compounds in the extract inhibited the proliferation of the HT-29 line. Antimicrobial properties were also investigated in the growth of L. rhamnosus, E. cloaceae, B. longum and B. adolescentis. However, this work did not show outstanding activities.
1.Line 18, human colon epithelial cell line CCD 841 841 shoule be CCD 841. The same as in line 349.
2.Line 26, double full stop.
3.Line 52, one more space in effects [6,7].
4.Line 107, following 2. Results, it should be 3.1.
5.Line 168, one more space 250 ug/mL.
6.Line 196, double Figure 5.
7.Lines 354, 360, and following, correct CO2.
8.Line 385, correct 1x105.
All changes have been applied as commented. This was an editing error.

Reviewer 3 Report
This work described the biological activity of an in vitro digested infusion of Epilobium angustifolium (fireweed). The content of selected phenolic compounds in the digested aqueous extract of fireweed was determined using LC-MS. Cytotoxicity was assessed by an MTT assay, a Neutral Red uptake assay, May-Grünwald-Giemsa staining, and a label-free Electric Cell-Substrate Impedance Sensing cytotoxicity assay. Although the work is very intersting, I think thare are some contents need to be modified.
- Eight compounds were selected for quantitative analysis. What was the basis for selection?
- Twenty six compounds were identified in the results. Please list the names of the 26 compounds in the manuscript.
- I suggest that the authors make a comparative analysis of the fingerprint of the extract before and after digestion.
Author Response
Thank you for the thorough review, which helped us to improve the publication. Authors have considered all Reviewers’ comments and corrected manuscript according to major and minor concerns.
Reviewer 3
This work described the biological activity of an in vitro digested infusion of Epilobium angustifolium (fireweed). The content of selected phenolic compounds in the digested aqueous extract of fireweed was determined using LC-MS. Cytotoxicity was assessed by an MTT assay, a Neutral Red uptake assay, May-Grünwald-Giemsa staining, and a label-free Electric Cell-Substrate Impedance Sensing cytotoxicity assay. Although the work is very intersting, I think thare are some contents need to be modified.
Eight compounds were selected for quantitative analysis. What was the basis for selection?
We selected these eight compounds on the basis of their amount in the extract. As we wrote in the manuscript P3L120 “The eight compounds listed in the table below were present in significant amounts.”
Twenty six compounds were identified in the results. Please list the names of the 26 compounds in the manuscript.
Twenty six compounds identified in the water extract of Epilobium angustifolium were as follows:
Citric acid, Gallic acid, Vanillic acid isomer, Galloyl-glucose, Digalloylglucose, Hydroxybenzoic acid isomer, Trigalloylglucose, Neochlorogenic acid, Hydroxybenzoic acid isomer, Coumaroyl-quinic acid isomers, Chlorogenic acid, Ferulic acid, Quercetin glucuronide, Vanillic acid isomer,Myricetin hexoside, Hyperin gallate, Myricetin glucuronide, Isoquercetin, Quercetin glucoside, Guaiaverin (quercetine arabinofuranoside), Kaempferol glucoside, Kaempferol glucuronide, Hydroxybenzoic acid isomer, Ferulic acid, Afzelin (kaempferol rhamnoside), Caffeic acid.
They have all been listed in the manuscript as suggested by the reviewer (P3L112-119).
I suggest that the authors make a comparative analysis of the fingerprint of the extract before and after digestion.
The aim of this study was to analyze the biological activities of fireweed infusion subjected to in vitro digestion, and not to analyze the chemical composition of the extract before and after digestion. We have already done such research and it was published in our previous work (Szwajgier et al. 2021, Evolution of the anticholinesterase, antioxidant, and anti-inflammatory activity of Epilobium angustifolium L. infusion during in vitro digestion. J. Funct. Foods 2021, 85, 104645, doi:10.1016/J.JFF.2021.104645) in which we analyzed the changes in the content of phenolic compounds (TPC, total flavonoid, total flavanols, condensed tannins and total anthocyanin) before, during and after in vitro digestion process (Supplementary materials). In this work, we focused on the eight compounds found in the digested fireweed infusion in the greatest amounts, as they probably determine the biological activity of the extract. In Table 2 in the manuscript, we present the percentage of these compounds in the E. angustifolium extract before and after in vitro digestion. Therefore, in the opinion of the authors, performing a comparative analysis of the fingerprint of the extract before and after digestion would be a repetition of the results from our previous published work and it is not necessary to achieve the assumptions and scientific goal of this work.

Round 2
Reviewer 1 Report
The authors has improved the manuscript substantially and incorporated most of the suggested changes and explained very well about the changes which they can not amend. The manuscript can be accepted after minor changes and authors must have to respond to these two points
To the query about figure 2 and 2, The authors have responded that that figure 1 is the cell viability of HT-29 and CCD 841 CoTr after 24 hours treatment with the extract while figure 2 shows the cell viability of extracts on same cell lines after 96 hours, but this must be reflected in the results and also in the caption of the figures. The authors need to amend the results and figures caption and clearly describe time dependent treatment.
2- Conclusion
Please amend the following sentence
The results we obtained show that a fireweed infusion can be used as a factor supporting the therapy of colon cancer, however in vivo or human tissues investigations should be performed to fully confirm this observation.
with this
"The results we obtained show that a fireweed infusion can be used as a factor supporting the therapy of colon cancer, however extract must be checked normal cells and in animals models in order to evaluate its toxicity".
Author Response
The authors has improved the manuscript substantially and incorporated most of the suggested changes and explained very well about the changes which they can not amend. The manuscript can be accepted after minor changes and authors must have to respond to these two points
To the query about figure 2 and 2, The authors have responded that that figure 1 is the cell viability of HT-29 and CCD 841 CoTr after 24 hours treatment with the extract while figure 2 shows the cell viability of extracts on same cell lines after 96 hours, but this must be reflected in the results and also in the caption of the figures. The authors need to amend the results and figures caption and clearly describe time dependent treatment.
Information on treatment time has been added to the Fig 1 and 2 captions and in the results section.
2- Conclusion
Please amend the following sentence
The results we obtained show that a fireweed infusion can be used as a factor supporting the therapy of colon cancer, however in vivo or human tissues investigations should be performed to fully confirm this observation.
with this
"The results we obtained show that a fireweed infusion can be used as a factor supporting the therapy of colon cancer, however extract must be checked normal cells and in animals models in order to evaluate its toxicity".
The sentence was changed as suggested by the Reviewer
Reviewer 2 Report
Dear Authors,
Please correct the following lists;
- Line 395, 3h;
- Line 431, 1.2 x 105;
- Line 535, 2017.
Author Response
Thank you for pointing out these three minor errors, they have all been corrected.
- Line 395, 3h; - is 3 h
- Line 431, 1.2 x 105; - is 1.2 x 105
- Line 535, 2017. - is 2017